# Separating Information on Transparent Polypropylene Labels on Packaging with Dual Properties for the Visible Spectrum and Instrumental Infrared Observation

**DOI:** 10.3390/polym14245341

**Published:** 2022-12-07

**Authors:** Denis Jurečić, Jana Žiljak Gršić, Diana Bratić, Silvio Plehati

**Affiliations:** 1Faculty of Graphic Arts, University of Zagreb, 10000 Zagreb, Croatia; 2Department of Informatics & Computing, Zagreb University of Applied Sciences, 10000 Zagreb, Croatia

**Keywords:** packaging, polypropylene label, V and Z information, IR graphics, NIR spectrum, security printing

## Abstract

Transparent linear NIR digital graphics with the intention of security information were printed on a polypropylene material. A label for expensive juices in transparent glass or plastic packaging is demonstrated. Obligatory information about the contents of the liquid was made to be read with the naked eye. The “Z” (near-infrared) information is expanded with data integrated for joint digital printing. This data does not disrupt the original, planned visual appearance of the label, “V” (visible). Although the two graphics are in the same place, the IR graphics on the label cannot be discerned with the naked eye. This brings elements of secrecy and protection against attempts to counterfeit the contents of the transparent packaging. The separation and recognition of the two pieces of information, V and Z, are achieved with a security camera in the near-infrared mode or with one of the many NIR detectors that surround us. In the article, the “VZ” algorithm for the integration of two independent graphics and the limitations set by digital printing are published. Toner properties and the method for developing the recipes for the composition of twin colorants for two spectral ranges are presented using spectroscopy.

## 1. Introduction

The widespread introduction of video surveillance in the NIR spectrum opens up the possibility to create labels for food products, which results in the exploration of colorant properties in two spectra. NIR cameras vary in type, from handheld detectors to cameras connected to computer databases [1].

In this article, the IRD (INFRAREDESIGN^®^) dual display of information on transparent polypropylene materials is demonstrated. The information on packaging is produced with colorants which absorb light differently in two ranges of the spectrum. In this article, the colorants are composed and programmed with the aim of absorbing and reflecting light in the visible and NIR spectrum according to the IRD procedure [2]. This solution helps to solve the problem of counterfeits by using a forensic approach.

It also helps the buyers of the product to verify the authenticity of the product by allowing them to see the hidden certificate [3]. Equipping the packaging with a polypropylene label that has graphics in two spectra means giving it an informational, aesthetic and protective code. The prototype polypropylene label is printed with a color laser printer with CMYK process colorants. The design of the label includes NIR hidden information that is visible with NIR detectors.

The process colorants C, M, Y and K differ in their properties when absorbing NIR radiation. The black tone of digital printing is assigned a new meaning and is achieved in two ways. The first way is as a mixture of equal coverages of cyan, magenta and yellow, called S. Since none of the C, M and Y colorants absorbs NIR radiation, the black mixture S does not absorb it either [4]. The second way is using the black colorant called K. The colorant K strongly absorbs NIR radiation. This is the basis of the idea of dual printing and of combining two independent graphics of which one will be invisible to the naked eye. It is detectable instrumentally with one of the NIR cameras.

The IRD procedure of replacing S and K colorants is our special version of the GCR procedure. Enhanced marking, which is the information in the NIR spectrum, is introduced. It is information hidden from our eyes.

The transparency of the polypropylene material and of the gap between the letters is the same before and after the two graphics are merged. The color of the apricot juice is visible through the transparent label on areas without the graphics and on planned areas in the NIR spectrum where there is no absorption of colorants.

Combining two graphics detectable in different ways is the direction of the development of a new technology in the field of forensics and specimens in security printing. The light blockade encompassed is in the range from 400 to 1000 nm [5]. The larger colored area of the polypropylene label requires a transparent design so that the contents of the packaging can be seen in the infrared region. The separation of information into two spectral areas offers quick detection of the originality of the polypropylene label. Products with similar polypropylene labels in the visible region can be easily distinguished by checking the label with dual design in the visible and NIR regions. This article gives the recipes for colorants for dual design used to achieve hidden information on a transparent material. Spectral analyses of colorant properties in two spectra have led to a series of studies on the duality of colorants in the visible and near-infrared spectrum [6]. Duality of colorants is tested on all kind of materials with different types of dyes and for different purposes, i.e., for enhancing urban security [7] and for fashionable clothing [8]. In this article, the application and solution of hidden information in the infrared spectrum is demonstrated on a polypropylene transparent label for expensive fruit juices in small batch production.

## 2. Packaging with VIS/NIR Duality Performed with IRD

A label for a fruit juice with dual information on two levels was designed. The first piece of information is intended for the visible spectrum. The second is intended to be photographed with a camera with filters for light blockades up to 1000 nm in the near-infrared spectrum [9].

Figure 1 shows the design of the label as our eyes see it. The beginning is in the RGB colors of our eye. The transition to process CMY colors is performed with the “none” version in GCR separation.

The V graphic has the same appearance as the one perceived by our eyes (Figure 1) before and after printing. It is the beginning for the IRD security label. The V graphic has a left side (V1 color image) and a right side (V2 black text) with information about the juice, which is obligatory on food packaging.

The Z graphic is monochrome in black tones. Figure 2 shows the graphic intended for the NIR spectrum. We call it Z. Detection will be performed using a camera and a corresponding filter that blocks the light of our visible spectrum. The Z graphic is designed as a text (APRICOT) in V1 left, while the V2 text ZAGREB, CROATIA belongs to the right side of the graphic in Figure 2. The textual information is recognizable, and such a proposal is accepted as security. Textual information is the best identification of the product. In our example of hiding security graphics in the infrared Z spectrum, a typography with extremely large letters is used.

The letters were screened with our “needle-like raster”. This black/white raster is not usual. A letter is interpreted as a pixel graphic with random vibrations of coverage with carbon gray color. This raster is the result of an experiment whose goal was to reduce the sensitivity of our eye in recognizing the point of contact of the black tone created with CMY colorants and the black colorant K for printing with toners for digital print.

We propose IRD (infrared design) as a new solution in the packaging technology. The idea comes from the field of the production of documents with a high level of protection against copying and reproducing. IRD guarantees the authenticity of food products.

After merging graphic V (Figure 1, RGB) and graphic Z (Figure 2, Gr), a CMYK graphic is achieved; it is primarily intended for the creation of a graphic for infrared printing. It is visually identical to the graphic in Figure 1. Its CMYK record is the same when observed with the naked eye as in Figure 1.

The difference in the arrangement of colorants (the transition from RGB to CMYK with infrared colorants) is completely different for areas V1 and V2, so the solution for printing is given at https://jana.ziljak.hr/apricot.pdf (accessed on 14 October 2022), where each of the channels C, M, Y and K can be separately analyzed. Colorant K is only in those elements of the image where the planned information is shown in Figure 2. This also shows the difference between GCR (gray component replacement) separation procedure and our VZ (VK) separation. The replacement of black V and Z tones is done only on a part of the letters (V1 and Z, Figure 2).

## 3. Separating the Visible from the Infrared in the Full Color Image

The designation “Z” is used for the initially set graphic that will be hidden from our eye. The designation “K” (carbon black toner) designates the hidden graphic only after the separation (prepress), that is, after printing. The difference between these two graphics is illustrated in Figure 3. Image Z, which determines the desired replacement of CMY with K, is introduced. It is a partial choice of the GCR (gray component replacement) procedure. The limitations of GCR, marked “VT” as the hidden information, create a new, unrepeatable security solution.

After merging the two graphics (Figure 1 and Figure 2), the letters will be recognizable but with partially lost and “damaged” values of coverage with black toner. Our eye does not see such a replacement of S and K. The graphic in Figure 1 has a left and right side. The left side is a graphic in color, and the result of merging V and Z graphics is partially shown in Figure 3. At the top of Figure 3 (Z), the letters “AP”, which are identical to Figure 2 (from the word APRICOT), are shown. After merging V1 (Figure 1, RGB) and Z (Figure 2) graphics, the state of the letters “AP” is shown as carbon black K (the bottom part of Figure 3). They are subject to possible (including maximal) replacements of C, M and Y with K.

The graphic “AP” is integrated into the color image (V1) with toner K. The other pixels are produced only with C, M and Y colorants. Our eye does not notice the letters “AP” integrated into the color graphic (the middle of Figure 3).

Its larger part is subject to the special GCR method we have named VZ or VK. The letters “AP” are hidden from view with the naked eye.

The letters within letters are shown separately. The letters in the part of the graphic called V2 (Figure 1) are colored with the colorants C, M and Y. We perceive them as a black tone. The letters have inner (transparent) gaps, both in the body of the letters, in the space between the lines and in the gaps between the letters.

In the middle of Figure 4 (CMYK) is the final solution CMYK, but only channel K, which can be seen in the light blockade at 1000 nm. Channel K is shown separately (Figure 4 (K)). At the top of Figure 4 (Z) is the initial graphic Z from Figure 2. It is not completely identical to the graphic in Figure 4 (K). The transition from state Z to state K is possible only in the places of the black bodies of letters (Figure 1). These letters (Figure 4 (CMYK)) were created from Figure 1 (V2) set as C, M and Y. That is why in an infrared graphic at 1000 nm there is only a graphic in channel K.

We perceive the whole of the initial text in graphic Z with our eyes. However, the letters are now produced not with three (CMY) colorants, but with four. The relation between the location of the text in the visible graphic (Figure 1, V) and in the Z graphic expands the topic of security printing. Even the smallest (mutual) shift in these locations would result in a different solution in channel K, that is, in the infrared graphics. This is important for the discussion and attempts to produce a copy, that is, for the topic of security graphics.

The initial text in graphic Z is experienced entirely with our eyes. Only the letters are now derived; not with three (CMY) dyes, but with four dyes (Table 1). The mutual position of the text in the visible graphic (Figure 1, V) and in the Z graphic expands the topic of security printing.

## 4. Separating Typography from the Visible to the Infrared

Copying or scanning the printed pattern of a label with dual graphics is not possible with the instruments of today. Scanners record the RGB state of images in the same way as photo cameras. The transformation and prepress are performed for CMY colorants with the GCR procedure. Several levels of colorant K insertion, i.e., replacement with carbon black toner, are available to graphic designers (none, mid, high and max). The VZ (VK) idea involves the input of colorant K only in the set amount Z and on the elements of the image designated as the independent graphic (Figure 2) intended for the NIR spectrum of observation. If the Z graphic is separated from the image when photographed with infrared reflectography, it would raise many questions about the merging of V and Z graphics. The IRR method is applied in museums and galleries only to inspect paintings and their (original) state in the NIR spectrum. The question remains how to merge two (RGB and Z-IRR) graphics in order to do the prepress for printing with VK duality.

We used text as a graphic symbol detectable with an IR security camera; it functions as the contours and body of the text depending on its shape and thickness. The symbol is planned and realized with 40% coverage of black colorant K, which is enough for our Z cameras. This work shows the possibility to detect originality of polypropylene labels in VIS/NIR spectral areas with cameras for detection and with cameras that can record images and videos for later forensics.

This article shows photos at 700 nm (Figure 5) and at 1000 nm (Figure 6). Figure 6 shows channel K, which is not identical to the initial Z graphic and is used in the beginning of the preparation of dual (VIS/NIR) graphics. It is known that more than a dozen products with the same name, but from different manufacturers, can be offered on the supermarket shelves.

Instrumental detection of the label was performed with Projectina PAG B50 [9], which allows 24 independent light blockades: N (daylight spectrum), 280, 385, 400, 420, 455, 475, 495, 515, 530, 550, 570, 590, 610, 630, 645, 665, 695, 715, 735, 780, 830, 850 and 1000 nm. We combined the forensic recordings in these blockades into an animation.

We propose animation as a new form of A-specimen presentation in the fight against counterfeiting. We talk about the specimen as proof of the originality of this kind of label on several levels: as partial photos in the light blockades and as a continuous transition from the VIS to the NIR spectrum. Through the animation (Figure 7) and by stopping it, you can see the gradual appearance of the graphic K until it is completely clearly displayed at 1000 nm (Figure 6). 

## 5. Discussion and Conclusions

Today, computer graphics is not only a field that studies algorithms for drawing or for programming the graphical interfaces. It now includes the creation, storage and processing of image content using computers, and we use these potentials to the maximum and with special creativity, continuously modernizing computer graphics both on the content and form.

By introducing the new variable Z to measure light absorbance at 1000 nm, our invention is implemented into computer graphics. The concept of “an image within an image” is shown in a completely new way. Today, the role of packaging is more diverse and complex than just preserving and protecting the product because it encompasses a much wider area. It can serve as a marketing tool and be the deciding factor in the purchase of a product; it can influence the consumer to act impulsively.

In our graphic design, as an interdisciplinary field combining typography, illustration, photography and, of course, printing, we also include the knowledge of infrared technology, therefore conveying more powerful ideas and deeper messages through visual communication. Thus, the reading of the image becomes not only on two levels, such as recognizing a photo and reading the given title of that photo, but also on three levels, because it is only at the hidden level that we can see what it is about and who the author of this complex work is.

A graphic with dual content, i.e., for the visible and for the infrared region, was also created for this article. It is a computer graphic for whose visible system V, an abstract composition composed of letters and color graphics of irregular geometric forms, was created. For the Z system, individualized textual information was embedded in the “image”.

The hidden text in the infrared graphic is a new solution that includes twins of process colorants for printing with a color laser printer. The IRD model supports the coverage of colorants, so the hidden text is separated in the range from white to almost black particles. 

There are two levels of presenting the merging of two graphics. The first is the combination of a colorful image in color with the letters “APRICOT” and originally screened (rasterized). The second is the combination of two black letter surfaces, where letter over letter with the reduction of colors is in such a way that the graphic is perceived as the same after merging and printing. Small letters carry information about the chemical composition of the apricot juice. In addition, the IR camera detects only the intended letters. Transparency is in the text: in the graphic with the words “ZAGREB”, “APRICOT” and “CROATIA”, the letters remain, but they have black Z properties only where the black letters of the text are made of small letters that describe the content and the obligatory information about the apricot juice.

## Figures and Tables

**Figure 1 polymers-14-05341-f001:**
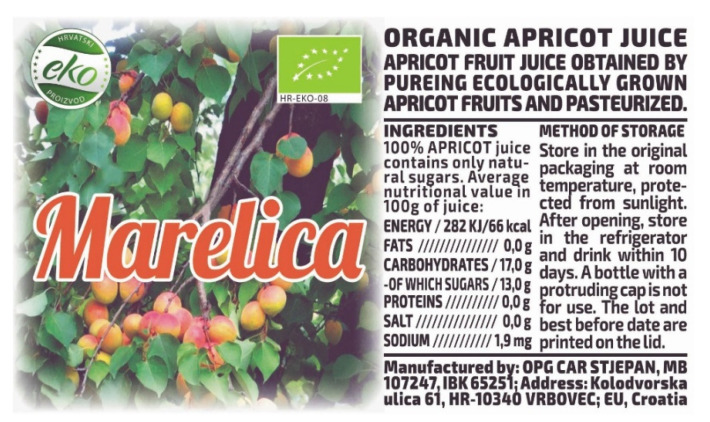
Original design “V” for the visible (RGB) spectrum.

**Figure 2 polymers-14-05341-f002:**
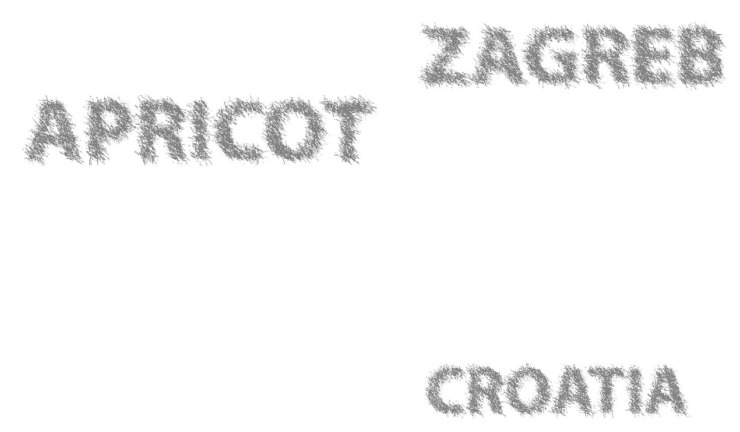
Gray graphic or “Z” intended for the NIR spectrum and planned as the invisible “Z” graphic.

**Figure 3 polymers-14-05341-f003:**
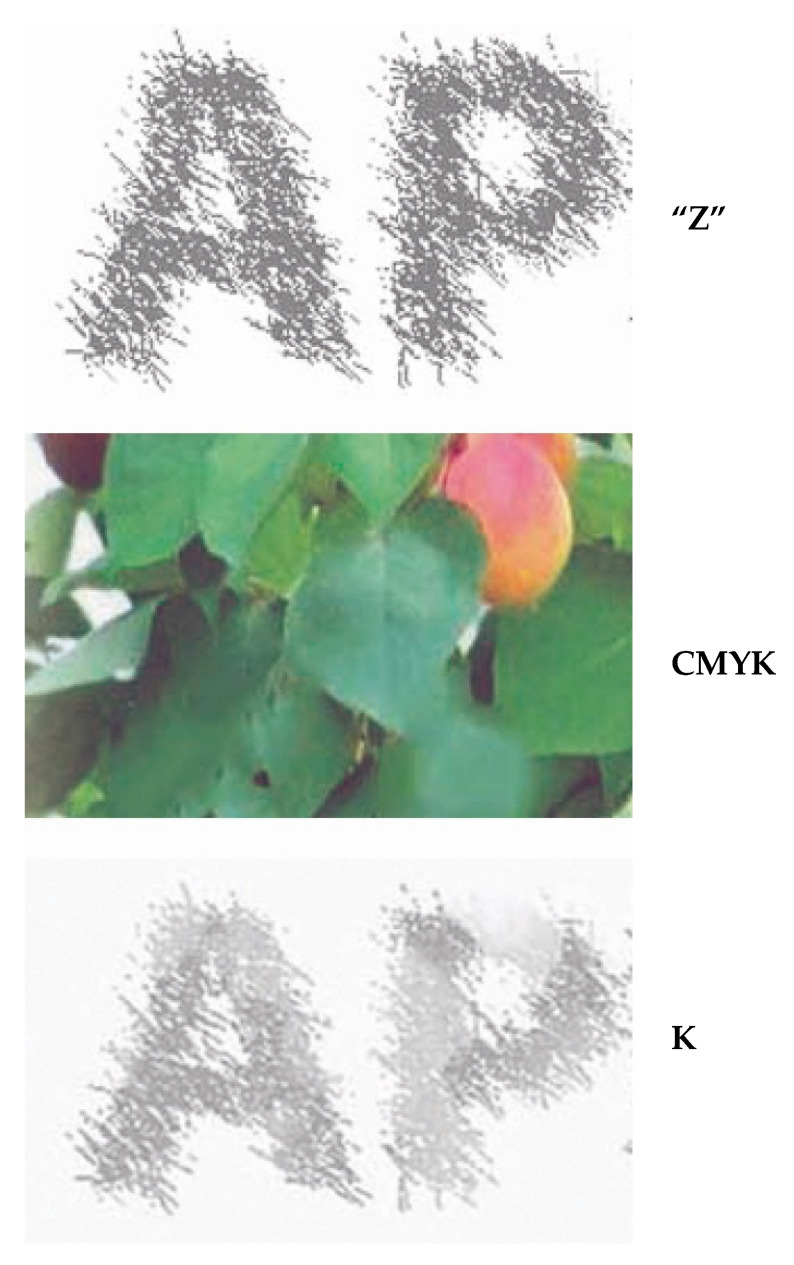
Hidden information in the color image.

**Figure 4 polymers-14-05341-f004:**
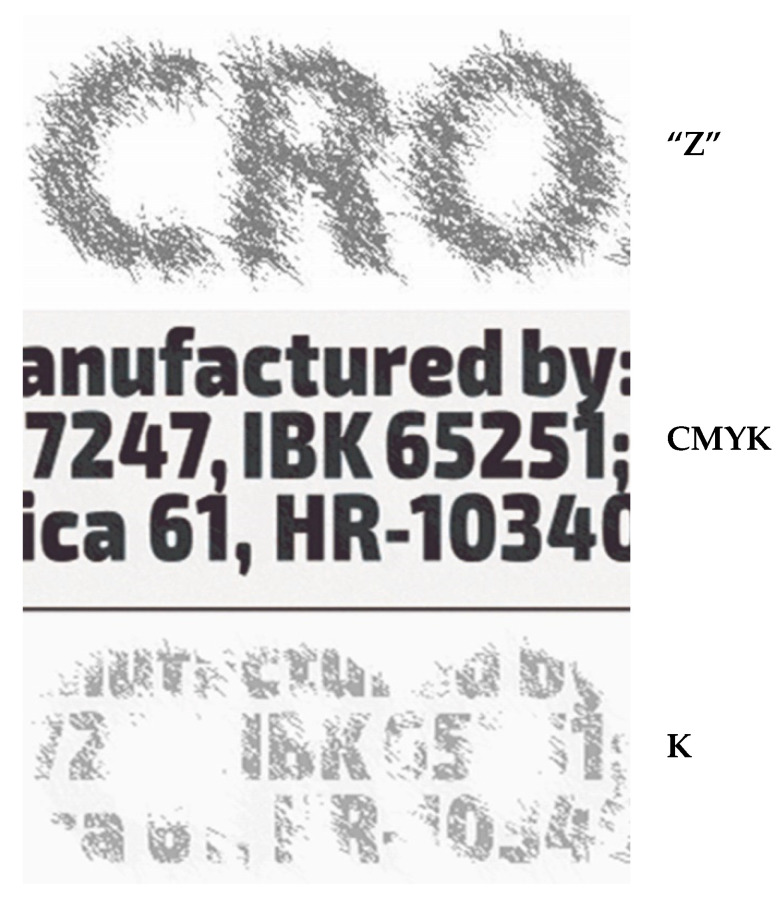
Letters within letters “CRO”.

**Figure 5 polymers-14-05341-f005:**
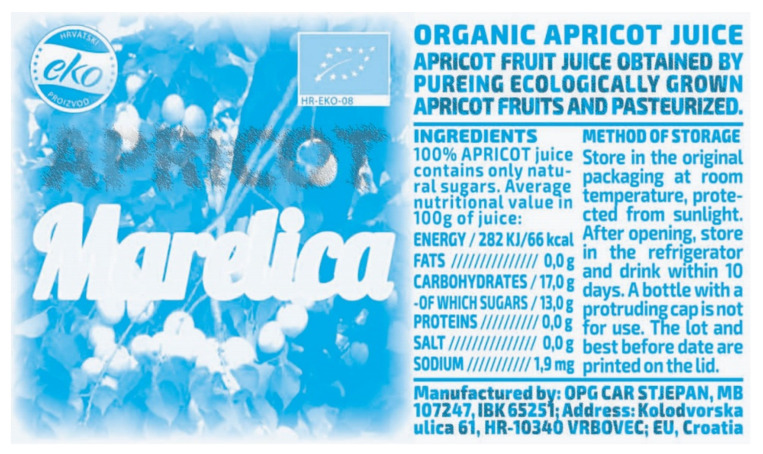
IRR photograph with a blockade at 700 nm.

**Figure 6 polymers-14-05341-f006:**
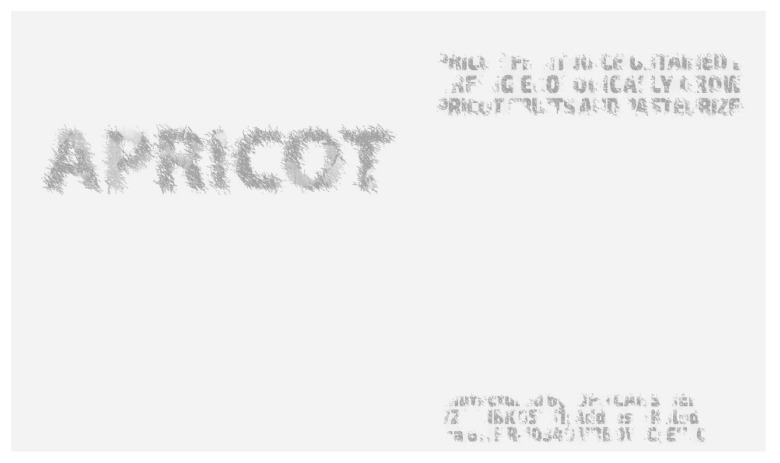
IRR photograph with a blockade at 1000 nm.

**Figure 7 polymers-14-05341-f007:**
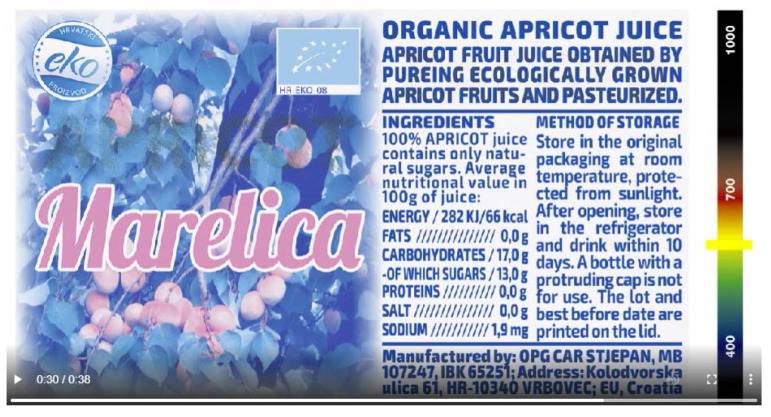
Animation of a specimen stopped with blockade at 600 nm: https://jana.ziljak.hr/apricot.mp4 (accessed on 14 October 2022).

**Table 1 polymers-14-05341-t001:** Linear graphics in 600 dpi with stochastics, raster elements, black letters and algorithm CMY. (*) asterisk indicates that it is a new color system (derived from the older one).

	Cyan	Magenta	Yellow	Black	L*a*b*
the initial state of the text	95	95	95	0	26, 2, 0
full tone of black (K)	75	65	60	40	27, −2, −3
transitional area of gray	55	35	30	40	41, −5, −8

## Data Availability

Data are contained within the article.

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
