# Peer review of "Separating Information on Transparent Polypropylene Labels on Packaging with Dual Properties for the Visible Spectrum and Instrumental Infrared Observation"

_polymers, 2022, doi:10.3390/polym14245341_

Round 1

Reviewer 1 Report

In this paper, the authors presented transparent linear NIR digital graphics for security information

on polypropylene material. The article is organized but needs some major improvements before to consider for publication.

1.       Please extend the introduction part by focusing on the relative reports published recently.

2.       Please add more relevant recently published references.

3.       What’s the importance of the recipes of the colorants of the dual design? Need to explain.

4.       Please explain what is the novelty here, and there is a need to perform such compact work. 

Reviewer 2 Report

The manuscript (polymers-2035682), Separating Information on Transparent Polypropylene Labels on Packaging with Dual Properties for the Visible Spectrum and Instrumental Infrared Observation, shows an interesting result for the visible spectrum and instrumental infrared observation. Authors presented quite comprehensive analysis in this manuscript. Some minor comment would like to provide here, shown as following - 

1. Please provide a simple benchmark Table to summarize the current research work and other research tasks. What kind of metrics authors have seen the improvement in here as compared to others? And why? This would be quite helpful for this referee and potential readers to get a further insight of the research work impact and contribution in this work as compared to other works. 

2. For the visible spectrum and instrumental infrared observation, does the area or scaling of the words/pictures/objects size impact here? What would be the major impact for objects scaling?

Due to the above comments, this referee would like to put the manuscript status as "Major Revision" in the current phase. 

Round 2

Reviewer 1 Report

Thanks for the revision

Reviewer 2 Report

Authors have replied to this referee in detail. No further comments from this referee.